# UNICONTACT: A Basic Model for Robotic Manipulation of Contact Synthesis on Rigid and Articulated Rigid Bodies with Arbitrary Manipulators

## Abstract

We posit that one fundamental, core component of robotic manipulation is inferring contacts with the environment, enabling the agent to exert control. In this work, we study a fundamental problem of contact synthesis in robotic manipulation to choose a set of contact positions and forces on a random rigid or articulated rigid object for an arbitrary robot manipulator to produce a specified external wrench. Our framework first segments the point clouds with normals into feasible contact region sets. For each feasible contact region set, a model is trained to produce the feasible contact point within these region sets by taking as inputs the robot description, the target wrench, the object point cloud with normals, and the contact region set. After gathering the contact positions from the neural network model, we develop an optimization process to fine-tune the contact points and contact forces and generate the joint values for the robotic manipulator to exert contact forces on the object's surface without penetration. We perform extensive experiments to verify the effectiveness of our proposed framework both in simulation and in real-world experiments. Supplementary and Videos are on the website https://sites.google.com/view/unicontact

## 1 Introduction

Empowering robots with the capability to manipulate objects of diverse shapes, types, sizes, and adeptly control their placements and status, has long been recognized as an essential component of intelligent robots Billard & Kragic (2019). Manipulation contains a wide range of subareas, including prehensile manipulation like grasping and reorienting pens, non-prehensile manipulation such as pushing coins and flipping a light switch, and rolling a ball on a table Bullock & Dollar (2011). Additionally, there are multipurpose fine manipulations such as tool manipulation and in-hand manipulation. According to Mason (2018), *manipulation refers to an agent's control of its environment through selective contact*. We posit that one fundamental, core component of robotic manipulation is inferring contacts with the environment, enabling the agent to exert control.

In this work, we formulate a fundamental problem of *contact synthesis* in robotic manipulation, which is to choose a set of contact positions and forces on a random rigid or articulated rigid object for an arbitrary robot manipulator to generate a specified target wrench. An effective and efficient robot model to generate contacts is crucial for a wide range of robotic manipulation applications. This is an extremely challenging problem due to many factors, such as the high complexity of object shapes, diverse connection patterns in articulated objects, and the large variety of robot designs.

This problem is related to several core issues in robotic manipulation, including *grasp synthesis*, *contact optimization*, and *affordance*. *Grasp synthesis* involves finding a grasp configuration that meets specific criteria, such as force closure to withstand arbitrary external wrenches. In contrast, our focus is on identifying contact points on objects to produce a target wrench. Our formulation is widely applied to non-prehensile tasks when robots need to manipulate non-graspable objects (for example, a large box without a grasping region). Furthermore, we extend the contact synthesis from rigid bodies to articulated rigid bodies, seamlessly integrating them into a unified framework.

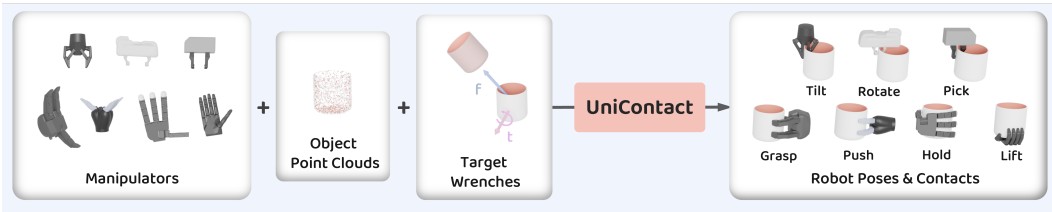

Figure 1: Given the object point clouds, the target wrenches, and arbitrary manipulator URDF, our proposed *UniContact* predicts the robot poses and contact points for different manipulation tasks.

*Contact optimization* typically assumes prior knowledge of object shapes and connectivity. Our framework, on the other hand, takes object point clouds as inputs, which could be gathered directly from the depth sensor. In addition to generating contact forces, our model contains an efficient pipeline to produce feasible, collision-free contact positions for specific robot configurations. Regarding *Visual affordances*, it describes the object manipulation process from the *robot-centric* to *object-centric* perspective. It highlights regions of interest on the object surface to indicate their pivotal role during the robot-object interaction. However, from the robot-centric view, the affordance regions are too coarse. In such cases, there may not be feasible motions to reach or manipulate these regions for particular robots. To achieve dexterous manipulation, robots must also determine the appropriate contact forces to exert on the objects.

Given the object point cloud, the target wrench, and the manipulator's URDF, our pipeline first segments the point cloud into multiple regions and infers these feasible region sets, which contain feasible solutions of the contact positions and forces. Then, the feasible contact regions, together with the manipulator's descriptions, and the target wrench, are fed into our proposed contact point generation network to produce the contact point sets and the associated robotic manipulator joint values, which serve as the initialization of our contact optimizations. Our proposed optimization generates accurate solutions for the robot to exert on the objects and provide a collision-free solution.

Our primary contributions are:

- We formulate a fundamental problem of *contact synthesis* in robotic manipulation for arbitrary manipulators to choose a set of contact positions and forces on a random rigid or articulated rigid object to generate a specified target wrench.
- We develop a neural network model to infer the feasible contact positions for arbitrary robot manipulators based on the manipulator's description, the target wrench, the object's oriented point cloud, and the contact region set.
- We propose an optimization framework to adaptively optimize contact force and contact location on the point cloud with normals.
- We develop an efficient collision-free inverse kinematic solver for robots to make contact with the specific positions on the object surface without further penetration, leveraging the artificial potential field.
- We conduct extensive experiments to verify the effectiveness of our proposed framework both in simulation and in real-world experiments.

## 2 PRELIMINARY

**Wrench**. When a force $f$ is applied at a specific point $p$ on an object, we can compute the resulting torque it generates around the origin, typically the center of mass. Then wrench, which combines both the force and torque, determines the acceleration exerted on the object by this force. At the point $p$ on the object surface, we can set up a local frame with $z$ along with the inward normal and $x$ and $y$ axis tangent to the surface. The mapping from force $f$ to wrench $w$ is linear and represented by the wrench basis matrix $G(p) = \begin{pmatrix} R \\ r \times R \end{pmatrix}$, where $R$ is the relative rotation matrix from origin frame to $p$'s local frame, $r$ is the position of $p$ and $r\times$ means the right cross product matrix of $r$. Then $w = G(p)f$.

**Fricion Cone**. When considering frictional contact force at $p$ on the object surface, we utilize the previous local frame of $p$. The contact force $f$ lies in the friction cone $FC = \{f \in R^3 : \sqrt{f_1^2 + f_2^2} \leq \mu f_3\}$, where $\mu$ denotes the friction coefficient. In practice, the friction cone can be approximated by a polyhedral cone spanned by a finite set of $k_f$ vectors $\{\hat{f}_i\}_{i=1}^{k_f}$.

## 3 TECHNICAL APPROACH

In robot manipulation tasks, the task description can be effectively represented by a wrench that we aim to apply to the object. For instance, the task "lifting an apple" translates to applying a straight upward force to the apple larger than its gravity, while the task "opening an oven" means applying a clockwise or anticlockwise torque along the axis of the oven door. Note that we assume that robots manipulate only one object/link at a time with the manipulator, whether it is free or articulated with other objects/links. The target wrench applied to that object/link is the only signal that we need for manipulation. The articulation information only influences which kind of wrench we should choose. Again in the task "opening an oven", the orientation of revolutional joint decides the direction of torque and the position of the joint decides the direction and magnitude of horizontal force so that the oven door can rotate along the joint axis. Thus the target wrench is enough.

Our framework takes the point cloud of a rigid body, the target wrench, and the robot URDF as inputs, and outputs the contact positions and contact forces on the object surface and the corresponding robot joint values to exert the forces at the positions. Note that our framework also requires point cloud's normals. Our pipeline first segments the point clouds. Denoted the segmented point cloud as $\{P_i\}_{i=1}^{\mathcal{S}}$. Based on the predefined number of contact points $\mathcal{K}$, our pipeline generates a list candidates of region set $\{P_j\}_{j=1}^{\mathcal{K}}$, where $P_j \in \{P_i\}_{i=1}^{\mathcal{S}}$. Here each contact region set contains $\mathcal{K}$ regions. Each region contains one contact point, thus one region may appear more than once inside the contact region set. Our pipeline select the contact points from the contact regions and produce the contact positions and forces, and the robot joint values.

The target wrench can be associated with the desired object pose and status. The discussion of how to specify the target wrench falls outside of our focus. We put the discussion to produce the target wrench in the website.

### 3.1 CONTACT-AWARE SEGMENTATION

For each point in the point cloud with its position $p_i$ and the associated inward normal $n_i$, we calculate each point's wrench basis matrix $w_i = [n_i, p_i \times n_i]$. We define each point contact feature as $c_i = [p_i, w_i]$ and cluster the point cloud, based on their contact features using Euclidean distance, into a finite number of regions which we call contact regions. Given a set of contact regions, we can immediately calculate if the set of contact regions is feasible to contain a set of contact points to produce the desired target wrench which is discussed later in this subsection. We apply the verification to all possible combinations of contact region sets. Each feasible region set is selected and fed into the deep network model to produce contact points (described in sec. 3.2).

Such a segmentation offers several benefits. First, the deepnet model described in the next subsection only needs to select the feasible contact points from a set of the contact region, which significantly reduces the learning difficulty. The segmentation also weights points according to their contact feature. Consider the task of opening the microwave, the point cloud of the front side of the microwave usually contains a large flat surface with a handle. Our pipeline segments these point clouds into several regions. The points of the flat surface contain one region, and the small surface on the handle becomes several regions. Although points located on the handle occupy a very small portion of the whole point cloud, their segmented contact regions are weighed equally to the region representing the large flat surface of the microwave.

Here we describe how to verify if the set of contact regions contains feasible contact points. Let $N_k$ denote the number of points in region $k$, the total point number of the set of $\mathcal{K}$ contact regions is $N = \sum_k^{\mathcal{K}} N_k$. We then consider the solution existence condition of target wrench $w$: equation $\sum_i^N G(p_i)f_i = w$ has non-zero solution $f_i$. From the preliminary, we know the friction $f$ is ap-

proximately a positive span of $\hat{f}_j$, where $j \in [1, k_f]$ and $k_f$ is the division degree of friction cone. So $w(p) = \sum_j^{k_f} a_j G(p) \hat{f}_j$ and $a_j \geq 0$. The target wrench equation becomes:

$$\sum_i^N \sum_j^{k_f} a_{i,j} G(p_i) \hat{f}_j = w, \ a_{i,j} \geq 0 \tag{1}$$

From convex analysis, we know if and only if the origin of wrench space lies in the convex hull of $\{G(p_i)\hat{f}_j\}_{i,j}^{N,k_f} \cup \{-w\}$, above equation has a non-zero solution. This condition helps to find out the feasible contact region sets.

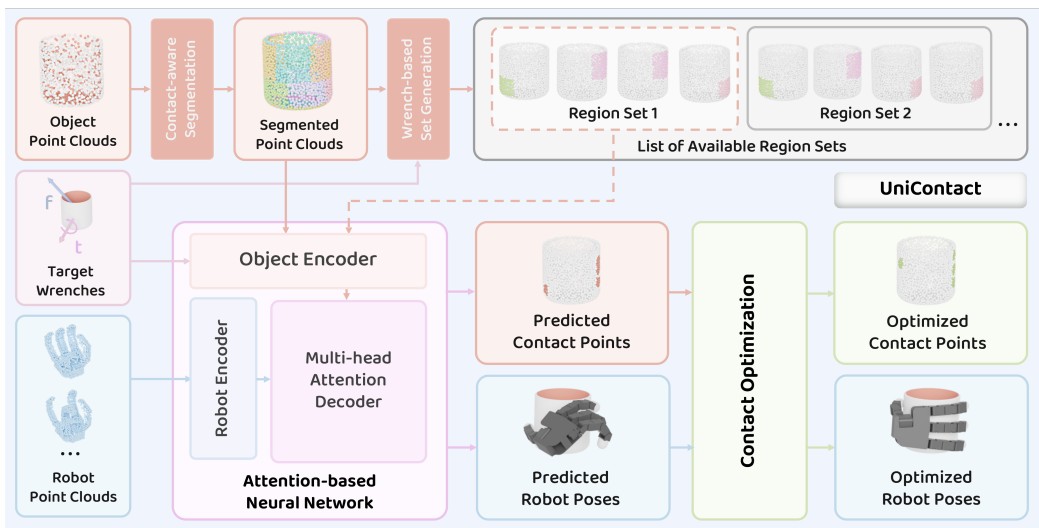

Figure 2: **Our proposed *UniContact* framework overview.** Given the target wrenches, we perform contact-aware segmentation on the object point clouds and generate a list of available region sets. An attention-based neural network takes in the region set, segmented point clouds, robot point clouds, and the target wrenches and outputs the predicted contact points and robot poses. Then we jointly optimize the contact points and robot pose based on the prediction to generate the target wrench.

### 3.2 CONTACT POINT SET GENERATION

Given a feasible contact region set, here we describe how to select the feasible contact point for the robot manipulator to generate the desired wrench. Specifically, the proposed network takes as inputs the robotic hand feature, the target wrench, the object point cloud, and the contact region set. If the contact region set has $\mathcal{K}$ contact regions, our network selects $\mathcal{K}$ contact points from the object point cloud. Each contact point is selected within its associated contact region. Note the contact region set might contain duplicate contact regions. The network architecture employs PointNet++ (Qi et al., 2017) as the encoder and constructs the decoder using the multi-head attention mechanism (Vaswani et al., 2017a). We first discuss the representation of the robotic hand and object and introduce the decoder for contact point selection. A more detailed network architecture is put at the project website.

**Robotic hand feature.** To describe the robot manipulator, we adopt the representation introduced in *UniGrasp* (Shao et al., 2020). We first learn a point-cloud auto-encoding network by training a neural network that takes as inputs the point clouds and reconstructs themself using Chamfer Distance (Fan et al., 2017). We adopt its encoder to generate a feature for each input point cloud, which can describe the robot manipulator's geometry at a specific joint configuration. Then, we want to describe the manipulator's joint range. We denote the upper and low limitations of joint $i$ as $H_i$ and $L_i$, respectively. Then, we have a middle configuration denoted as $(M_1, ..., M_n)$, where $M_i = \frac{1}{2}(H_i + L_i)$. Each joint takes its upper or low limitation, while others take its middle values of the joint range. In this way, there are $2n + 1$ joint configurations in total, e.g.

$(H_1, M_2, ..., M_n), (L_i, M_2, ..., M_n), (M_i, H_2, ..., M_n)$. The point clouds are fed into the encoder of the auto-encoding network to get $2n + 1$ global features, which later are concatenated and combined to produce a final robot manipulator feature.

**Object encoder.** We adopt *PointNet++* to extract object features from the point clouds. The network encoder takes as inputs object point clouds and point clouds' normal vectors, target wrench $w \in \mathcal{R}^6$ and the contact point region set masks (similar to segmentation masks). Note that the target wrench vector is repeated by the number of point clouds and directly added to each point in the point cloud, along with each point's 3D position and normal.

**Point selection decoder.** The point selection process is modeled as a seq2seq task. Inspired by Transformer (Vaswani et al., 2017b), we propose a point selection decoder with a multi-head attention mechanism for the decoder. For a region set containing $K$ regions, we sequentially feed the object point clouds together with the corresponding contact region mask. The $i$th iteration, we have the contact region mask denoted $\mathcal{M}_i$, where $i \in [1, \mathcal{K}]$. For points in the $i$th contact region, their $\mathcal{M}_i$ is one. Otherwise, their $\mathcal{M}_i$ are zero. The mask with the point clouds is fed into the decoder to extract the object feature. Additionally, the robotic manipulator feature is also fed into the decoder to guide the neural network model to select contact points suitable for the robot manipulator. Under the attention network, the object feature plays as the query while the robot manipulator feature plays as the key-value. The neural network selects a contact point within the contact region masked by $\mathcal{M}_i$ as the output.

Due to the page limit, we put the network's training details, the loss function, and testing descriptions in the project website.

### 3.3 Optimize on Selected Region Set

The deepnet model outputs the positions of a contact point set. However, the neural network suffers from an estimation error, and we need to generate executable robot actions. Thus, we develop a fine-tuning module to optimize the contact positions and forces to precisely match the target wrench. Meanwhile, it produces the robot's joint values to manipulate the object through these contact points.

The fine-tuning module contains several iterations. Each iteration consists of 3 stages: 1) fix the contact positions, optimize the contact force to match the target wrench, 2) jointly optimize contact positions and forces to move the contact points in their neighborhood to search for better contact positions, 3) solving collision-free inverse kinematics (IK) to generate new joint values for robots to move to new contact points. If there is no feasible IK solution, we reject the current contact point sets and ask the neural network to produce another contact point set.

**Contact Force Optimization** The contact force optimization problem (CFP) optimizes the contact forces $f_i$ by minimizing the difference between the contact wrench and target wrench $w$. Unlike common force optimization problem Boyd & Wegbreit (2007), which assumes the problem is feasible and minimizes the magnitude of contact forces subject to target wrench equation $\sum_i G(p_i) f_i = w$.

Our problem does not guarantee feasible solutions at these given contact positions. Thus we minimize the differences between the wrench introduced by contact forces and the target wrench $w$.

$$\min_{f_i} \left\| \sum_i G(p_i) f_i - w \right\|, \text{ s.t. } f_i \in FC \tag{2}$$

If the difference between these two wrenches is less than a given threshold, these contact points are considered feasible to produce the target wrench, and we can use a common force optimization Boyd & Wegbreit (2007) to minimize the magnitude of forces. Otherwise, we optimize the contact positions to find better contact points described below.

**Jointly Contact Position and Force Optimization**. The previous contact force optimization 2 returns the optimal contact forces $f^*$, which can be regarded as a function of contact points $f^*(p)$. So our optimization objective $\| \sum_i G(p_i) f_i^*(p) - w \|$ is an objective function of contact positions $(p_1, ..., p_\mathcal{K}) \in S^\mathcal{K}$, where $S$ denotes the surface of object and $\mathcal{K}$ denotes the number of contact points

within a set. The neighborhood of any contact point can be linearized to the surface's tangent space at that point. Thus, the optimization variables are contact point movements $\delta p_i \in R^2$ and contact force differences $\delta f_i \in R^3$. We Taylor expand the grasp mapping matrix $G(p)f^* = G(p^*)f^* + \nabla G(p^*)\delta p f^* + G(p^*)\delta f + \nabla G(p^*)\delta p \delta f$ and ignore second order terms of $\delta f$ and $\delta p$. Through this linearization, we rewrite the optimization objective as a linear function of contact positions, which is a convex optimization shown below and can be solved easily.

$$\min_{\delta f_i, \delta p_i} \left\| \sum_i \Big\{ \big(G(p_i^*) + \nabla G(p_i^*)\delta p_i\big) f_i^* + G(p_i^*)\delta f_i \Big\} - w \right\| \tag{3}$$
$$\text{s.t. } f_i^* + \delta f_i \in FC, \ \|\delta p_i\| \leq s$$

Here $\delta p_i$ is bounded by a circle of radius $s$, where $s$ means the max move step of contacts, and $f_i^* + \delta f_i$ lies in the friction cone, so both constraints are convex. The derivation of $\nabla G(p^*)$ is put in the supplementary.

**Collision-free Inverse Kinematics** Given the updated contact positions, our framework calls an inverse kinematics solution to generate collision-free joint values for the robots to precisely make contact at these specific points on the object surface while avoiding further penetration. Detailed procedure is described in next subsection 3.4.

**Error Tolerance** If we set target wrench $w$ to zero and have the residual of CFP $\|\sum_i G(p_i)f_i\|$ converges to zero. This means the grasp nearly forms a force closure, as long as every $f$ is in the interior of the friction cone. In order to ensure this condition, we can replace the current friction cone with a smaller convex set in the optimization. For example, we can choose a smaller friction coefficient $\mu$ and add a new constraint that each normal force should be larger than a threshold $f_3 \geq \eta$. This new convex set is a proper subset of the original friction cone. So if the residual converges to zero under such constraints, every $f$ is in the interior of the friction cone and the grasp forms a force-closure. If $w$ is nonzero, the optimized grasp no longer needs to form a force-closure, but using stronger constraints still helps to improve the error tolerance of the optimized grasp. The smaller $mu$ and the larger $\eta$ we use throughout optimization, the more stable grasp we will obtain after optimization.

### 3.4 Collision-Free Inverse Kinematics

Within each iteration of adjusting the positions of contact points, the robot must change its fingers to precisely attach to contact points but without object penetration. This challenging inverse kinematics (IK) problem proves difficult for current IK solvers like Ranged-IK (Wang et al., 2023) and IKFast(Diankov, 2010), which struggle to balance intersection-free conditions with reaching target positions. To address this, we propose a two-stage IK Solver, leveraging the artificial potential field (Khatib, 1985) to generate collision-free finger positions in contact with the specific points while avoiding further penetration.

**Stage One: IK without collision avoidance** In the first stage, we ignore the existence of objects and minimize the distances between fingertips $t_i$ and contact points' position $p_i$. We use the gradient descent method leveraging the differentiable physic simulation (Yang et al., 2023), but any standard IK solver suffices.

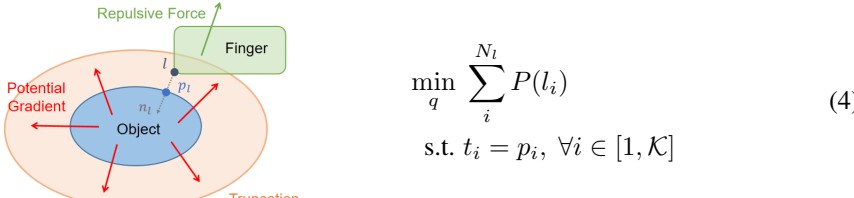

$$\min_q \sum_i^{N_l} P(l_i) \tag{4}$$
$$\text{s.t. } t_i = p_i, \ \forall i \in [1, \mathcal{K}]$$

**Stage Two: IK with artificial potential fields** After stage one, the fingertips have reached the specific contact points' position. Then, we attach the fingertips at the specified contact positions (similar to ball joints) and let other robot links move freely in the artificial potential field. We define the potential field similar to the truncated signed distance function (SDF), which pushes robot links

away from the object surface as shown in the following Fig **??**. Let $\mathcal{K}$ denote the number of locked fingertips and $N_l$ denote the number of other robot links. Given the position of a link $l_i$, the potential is $P(l_i) = n_{l_i} \cdot (l_i - p_{l_i})$, where $p_{l_i}$ is the projection point of $l_i$ onto the object surface and $n_{l_i}$ is the inward normal at $p_{l_i}$. The problem then forms an optimization:

In Eqn. 4 $t_i = p_i$ indicates the $i$th fingertip is locked to the position $p_i$. The loss function is the sum of potentials over all robot links except finger tips that are attached to the surface, $L_p = \sum_i^{N_l} P(l_i)$. Its gradient is $\nabla_q L_p = \sum_i^{N_l} n_{l_i}^T J_{l_i}$, where $J_{l_i}$ is the $i$-th robot link's translation Jacobian. Considering the constraints that $N_t$ finger tips are locked, the update value $\delta q$ should satisfy $J_t \delta q = 0$, where $J_t = [J_{t_1}^T, ..., J_{t_{N_t}}^T]^T$ and $J_{t_j}$ is the $j$-th finger tip's translation Jacobian. We then construct $\delta q$ from the gradient $\nabla_q L_p$ by minusing its projection onto the constraint: $\delta q = -\nabla_q L_p + J_t^+ J_t \nabla_q L_p$, where $J_t^+$ is the Moore–Penrose inverse of $J_t$ that $J_t J_t^+ J_t = J_t$. It's easy to verify $J_t \delta q = 0$. So the joint values are iteratively updated by $q \rightarrow q + \lambda \delta q$ until every $P(l_i)$ is below a negative threshold, where $\lambda$ is the learning rate. An illustration figure is available on the project website. With these two stages, our IK solver returns the solved joint values.

## 4 DATASET

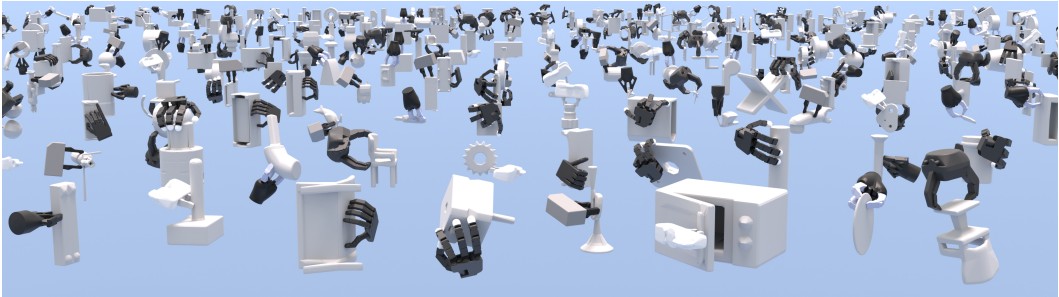

Figure 3: *UniContactNet* **Dataset Visualization.** Millions of contact training examples with seven manipulators and 100K+ rigid objects.

**Object Pre-processing** We collect 100K+ object models from 1K+ categories in Objaverse (Deitke et al., 2022), ShapeNet (Chang et al., 2015), ABC (Koch et al., 2019), Thingi10K (Zhou & Jacobson, 2016), and GAPartNet (Geng et al., 2023). We perform filtering, remeshing into manifolds (Huang et al., 2020), rescaling, aligning the center of mass with the coordinate origin, and convex decompostion (Wei et al., 2022) on the object models.

**Annotations** Our dataset contains millions of training examples. Each example is composed of: object point cloud $\mathbf{p}_O$, object point cloud normals $\mathbf{n}_O$, contact-aware segmentation $\{P_i\}_{i=1}^S$, a hand palm pose $^O X^H$, joint configuration $q$, contact points $\mathbf{p}_c$, finger torques, arm force, contact forces $\mathbf{f}_c$, and wrench $w$. We propose a unified sampling-based method for large-scale contact synthesis. We adopted a hierarchical sampling approach to obtain suitable contact training examples. For a scaled object model, we sample points on its outer surface and obtain their corresponding normals. Since solving the inverse kinematics (IK) for multiple fingers with a floating base is relatively challenging and time consuming, we propose first sample palm poses facing the object, solve IK for each finger, and perform combinations on all finger solutions. Under the joint configuration, we sample joint torques and arm-to-palm forces and find combinations to provide contact forces within all the contact friction cones. At last, we calculate the sum of the wrenches exerted by all fingers to the object $w$. Here we focus on point contact between the tips of robotic hands and the object surface under the quasi-static assumption.

## 5 EXPERIMENTS

In this section, we conduct experiments to answer the following questions:

Q1: How does our proposed neural network model compare to other approaches?

|  | Panda Hand | | | Kinova3F Hand | | | Allegro Hand | | | M-Allegro Hand | | |
|---|---|---|---|---|---|---|---|---|---|---|---|---|
|  | SR | OT | OD | SR | OT | OD | SR | OT | OD | SR | OT | OD |
| UG | 0.87 | 0.089 | 0.100 | 0.75 | 0.080 | 0.122 | 0.70 | 0.035 | 0.138 | 0.87 | 0.046 | 0.164 |
| **Ours** | **0.87** | **0.034** | **0.046** | **0.87** | **0.044** | **0.053** | **0.96** | **0.016** | **0.093** | **0.97** | **0.010** | **0.088** |

Table 1: Comparative Analysis of Contact Point Quality. The experiments are conducted with UniGrasp (UG) and UniContact (Ours) for four different robotic hands. M-Allegro Hand refers to Allegro Hand with no ring finger and this hand is unseen in the training dataset.

Q2: How effective is our proposed fine-tuning module to update the contact positions and forces?

Q3: How effective is our proposed collision-free inverse kinematics solver?

Q4: Is our proposed approach sensitive to the noises and perturbations? (See website)

Q5: How effective is our proposed model of taking the contact region sets compared to directly taking the whole point cloud? (See website)

Q6: What are the performances of our proposed framework on real point clouds and real robot experiments? (See website)

## 5.1 UNICONTACT EFFICACY: COMPARATIVE ANALYSIS

**Baseline comparisons on point selection framework.** We compare our network model with *UniGrasp* (Shao et al., 2020), a multi-stage model designed to select contact points in force closure. Our setting's annotations aren't a good fit for UniGrasp. However, to the best of our knowledge, UniGrasp has the most similar input, output, and setting with us. Therefore, we conduct a comparison experiment with UniGrasp. The results are reported in Tab 1.

We totally use three metrics in the evaluation. **SR** stands for the success rate of finding contact points after the optimization (described in Sec 3.3) to successfully generate the desired target wrench (within a given small difference). Higher values are better for this metric and it shows the final performance for manipulation. While **OT** and **OD** refer to the time for optimization and distance between initial contact points and those after optimization, respectively. Lower values are better for these two metrics and they measure the initialization quality. As is shown in Tab 1, our method outperforms UniGrasp on all the metrics. The higher success rate means that our network model along with optimization can successfully accomplish the manipulation task. The lower OT and OD validate that the network model provides the optimization with good initialization.

**Evaluation of the generalization to Novel Manipulator** We edit the URDF of the allgro hand and delete the little finger, making it a new three-fingered gripper. We feed the new gripper into our pipeline and report the performances denoted as *M-Allegro Hand* in Tab. 1. The result indicates that our model remains to produce reasonable results for this new hand.

**Evaluation of the fine-tuning module** To compare the contact points quality of the positions directly returned by the neural network and the positions returned after the fine-tuning module, we report how close the two wrenches induced by these contact positions to the target wrench. The wrench error defined in Eqn. 2 is reduced from 0.0165 (before optimization) to below 0.0001 (after optimization).

**Evaluation of the IK Solver** We visualize the inverse kinematics results described in Sec. 3.4 after the **Stage One** and **Stage Two**. The allegro hand is pushing a cup through two contact points. In stage One, the hand intersects through the cup, while in Stage One, the two fingers are precisely in contact with the cup's surface. More quantitive and quantitive results are put on the website.

## 5.2 REAL-ROBOT EXPERIMENTS

We set up the real world experiments with a dual-arm robot as shown in Fig 5.1. The MOVO robot Kinova has two 7 DoF arms and a Kinect RGB-D camera over head. We gather the segmented point cloud leveraging the image segmentation from SAM Kirillov et al. (2023). We test our pro-

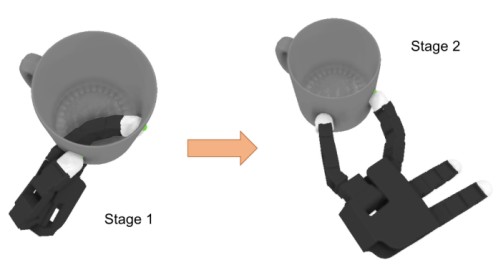
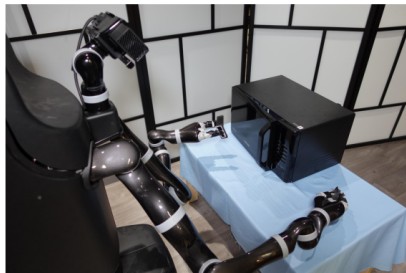

(a) IK with Artificial Potential Field                    (b) Real-Robot Experiment

posed approach both on rigid bodies such as the bottle, cup, and pan, also on articulated rigid bodies such as microwave. Videos are available at our project website

## 6    RELATED WORK

**Grasp synthesis** studies the problem of finding a grasp configuration that meets a set of criteria relevant to the grasping task. For a broader review, we refer to Bohg et al. (2013). Shao et al. (2020) developed the *UniGrasp* to generate contact points for multi-fingered robotic hands to grasp random objects. *AdaGrasp* (Xu et al., 2021) learn a single policy to generate grasp poses that generalize to novel grippers. Recently, there has been increasing interest in multi-fingered grasping (Liu et al., 2023; Xu et al., 2023; Wan et al., 2023). Our proposed model contains the grasp synthesis but is not limited only to grasping. Our models contain many other manipulations, such as pushing, rotating, and pulling.

**Contact Optimization** There is a rich literature about contact optimization for robotic manipulation with various optimization objectives and optimization strategies. Hang et al. (2016) proposed the hierarchical Fingertip Space (HFTS) as a representation enabling optimization for both efficient grasp synthesis and online finger gaiting. Turpin et al. (2022) leverage the differentiable simulation to generate grasp. Fan et al. (2018) Contact-implicit trajectory optimization (CITO) Mordatch et al. (2012); Marcucci et al. (2017); Cheng et al. (2021); Gabiccini et al. (2018) plan manipulation actions without a pre-specified contact schedule. Zhu et al. (2023) propose a high-level finger gaiting scheme and utilizes differentiable physics simulations (Yang et al., 2023) for contact optimization and contact localization for efficient search.

**Learning Visual Affordances** (Do et al., 2018; Yen-Chen et al., 2020; Lin et al., 2023) has received increasing attention for robotic manipulation, including grasping (Mandikal & Grauman, 2021; Borja-Diaz et al., 2022; Wu et al., 2023) and articulated object manipulation (Mo et al., 2021; Wu et al., 2021; Wang et al., 2022). For a broader review, we refer to Hassanin et al. (2021). Mo et al. (2021) proposed Where2Act to predict the actionable point-on-point cloud for primitive actions such as pushing and pulling. Borja-Diaz et al. (2022) These affordances are usually specified for particular manipulation tasks. Our proposed framework.

## 7    CONCLUSION

In this work, we propose a contact synthesis framework *UniContact* to generate contact positions and forces for arbitrary robotic manipulator to manipulate random rigid and articulated rigid bodides. Given the object point cloud, the target wrench, the manipulator's URDF, our pipeline first segments the point cloud into multiple regions and infers these feasbile region sets which contains feasible solutions of the contact positions and forces. Then the feasible contact regions together with manipulator's descriptions, the target wrench are fed into our propose contact point generation network to produce the contact point sets and the associated robotic manipulator joint values, which serves as the initialization of our contact optimizations. Our proposed optimization generates an accurate solutions for the robot to exerts on the objects and provide a collision-free solution. We conduct extensive experiments to verify the effectiveness of our proposed framework both in simulation and in the real-world experiments.

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
