# OpenReview forum: "UniContact:A Basic Model for Robotic Manipulation of Contact Synthesis on Rigid and Articulated Rigid Bodies with Arbitrary Manipulators"
_ICLR.cc/2024/Conference — Submitted to ICLR 2024_

### Official Review · Reviewer_UfwH · 2023-10-23

**Soundness:** 2 fair
**Presentation:** 1 poor
**Contribution:** 2 fair
**Rating:** 3
**Confidence:** 3

**Summary:**

This paper focuses on generating gripper configurations using object point clouds, with a significant contribution being the optimization of the chosen grasp area to create collision-free gripper postures. The research notably includes tests across multiple gripper types, demonstrating versatility in applicability, an advancement in the robotics and computer vision fields.

**Strengths:**

This paper employs optimization techniques for contact and collision avoidance, enhancing the system's overall framework and presenting a clear and comprehensive logic.

**Weaknesses:**

The paper lacks sufficient comparative experiments, emphasizing the complexity of robotic hand inverse kinematics. However, recent studies have proliferated in generating desired human hand postures based on object models. There is a certain similarity between the collision-free generation of robotic gripper configurations on objects and that of human hands, which calls for comparative experiments to demonstrate the superiority of the proposed method.

Moreover, concerning novelty, the paper employs the Artificial Potential Field method, a classic collision avoidance algorithm, though there are more advanced methods that might perform better. This traditional approach, while reliable, is not necessarily at the forefront of current technological advancements in robotics.

The author could strengthen the paper by comparing the proposed method with contemporary strategies that also aim for collision-free grasping. This comparison could highlight specific advantages in efficiency, accuracy, or applicability in diverse scenarios.

Additionally, while the paper highlights the intricacies of inverse kinematics in robotic grasping, a more in-depth exploration and comparison with human-like grasping techniques are advisable. These comparisons could offer insights into natural and intuitive grasping postures, potentially improving the robotic system's performance and versatility. It would be compelling to see if incorporating advanced techniques could further optimize collision avoidance and gripping efficiency, making a stronger case for the proposed method's applicability and superiority.

**Questions:**

The concerns you raised highlight significant areas in the research presentation that need improvement. First, the inability to access supplementary materials like the experimental demos, detailed network information, data, and video demonstrations that the paper references is a major drawback. These materials are often crucial for readers to fully understand, replicate, or even extend the research, and their absence can limit the paper's impact and credibility.

1. **Inaccessibility of Supplementary Materials**: The authors should ensure that all supplementary materials referenced in the paper are readily accessible. This may require updating the paper with working links or providing an alternative means of access, such as a supplementary appendix or a stable public repository. This accessibility is paramount, especially for readers and researchers who rely on these resources to deepen their understanding or build upon the existing work.

2. **Incomplete Sections**: Regarding the observation that the paper appears unfinished, particularly in the section on related work, this is a critical issue. The related work section is fundamental in any research paper as it situates the research within the context of existing literature, highlighting the unique contributions of the paper and building its premise on previously established concepts, techniques, and findings.

   - The authors must address any areas of the paper that appear incomplete by providing a comprehensive review of relevant literature, discussing how their work is different from and/or improves upon previous approaches. This is not only important for situating the research in its academic context but also for justifying the paper's contributions.

In conclusion, the authors need to address these significant shortcomings by ensuring the complete accessibility of supplementary resources and completing all sections of the paper thoroughly. These steps are necessary to enhance the paper's reliability, comprehensiveness, and overall contribution to the field.

**Details Of Ethics Concerns:**

No ethics review is needed

---

### Official Review · Reviewer_1VGY · 2023-10-31

**Soundness:** 2 fair
**Presentation:** 3 good
**Contribution:** 2 fair
**Rating:** 5
**Confidence:** 3

**Summary:**

This work proposes UniContact framework to predict contact points and robot poses for arbitrary robotic manipulator to grasp rigid objects. The work extends the UniGrasp to concurrently predict both the contact points and robot (hand) poses, given by the more input of wrench.

**Strengths:**

1, This paper formulates the robotic grasping task as applying a wrench to the object, which is more physically intuitive. Given object point clouds, target wrenches and robot point clouds, the framework can output contact and robot poses directly.

2, The training data includes millions of training examples with diverse 100K+ object models from 1K+ categories and different kinds of manipulators. UniContact can produce valid contact point sets not only on novel objects but also generalizes to new robotic manipulator.

**Weaknesses:**

1, In inference stage, how can users obtain the target wrench of an object with specific task description? For the same object but different task requirement, the desired wrench might be different. The efforts to generate the target wrench should be discussed further in the paper to evaluate whether is realistic for robot manipulation.

2, It is not clear how this new formulation benefits autonomous robot manipulation. I have doubt that the delicate calculation of required wrench as in input for a manipulation task may make the task complex, which contradicts the purpose of learning-based approach.

3, Another concern about the paper is its technical novelty. The paper is mainly inspired by UniGrasp. As much as I appreciate the novel improvements including contact optimization and IK solver, I had a hard time justifying the technical novelty of this paper given that the main focus of the ICLR conference is on learning methods themselves.

4, The experiment part is weak and insufficient, where the baseline is only UniGrasp and there is no ablation study or detail analysis provided.

5, Some key information and experiments are stated in the manuscript to be provided in the website link. But until now, the website is still empty. With these, it makes the paper hard to understand.

6, There are some grammar errors in the paper, including the figure index in the first line of 7th page, the last sentence in RELATED WORK etc.

**Questions:**

1, I also have doubts about whether to generate the required wrench by hand kinematics and pose is meaningful. For example, to lift up the object, the hand/gripper can be just to exert the static friction force on the object; any force, motion speed can be externally applied by the robot instead of the hand/gripper? The concept of wrench will be only meaningful if considering the dynamics of object.

2, UniContact seems only focus on single object grasping. Can it also apply to more realistic task such multiple/stacked objects grasping in cluttered environment?

---

### Official Review · Reviewer_G6eg · 2023-10-31

**Soundness:** 2 fair
**Presentation:** 3 good
**Contribution:** 2 fair
**Rating:** 3
**Confidence:** 3

**Summary:**

This submission introduces a pipeline for contact synthesis in robot manipulation scenarios. Given a point cloud of the target object, along with a geometric representation of the robot’s end-effector and a description of the target manipulation task (in the form of desired linear and angular acceleration in 6 DoF (wrench)), the proposed pipeline identifies target contact positions and required forces (through a learnable approach) that are subsequently translated to robot joint positions via a two-stage inverse kinematic solution that ensures collision-free manipulation.  A large scale dataset of manipulation contacts with synthetic objects, featuring a large variety of different objects and robot end-effectors is also contributed.

**Strengths:**

-The examined manipulation problem is very ambitious and the provided solution contributes towards generic robot skills, in the sense that the proposed approach is aimed to be applicable on arbitrary robot end-effectors and is able to deal with previously unseen objects.

-The provided problem formulation and contributed dataset can facilitate further research in the field.

-Overall the manuscript is well-written and easy to follow, and provides adequate context for non-expert readers.

**Weaknesses:**

The proposed methodology makes some severly constraining assumptions:

-The examined setting seems to assume that no external forces are applied to the manipulated object at any time, other than the manipulation forces from the robot. In the vast majority of real-world manipulation tasks this assumption does not hold, as friction and gravity play an important role on the required manipulation forces (for complex tasks other than grasping, where the submissions aims to focus).

-The proposed Inverse Kinematic solution does not consider collisions with background (support) objects (e.g. a table), or clutter from other objects in the scene. This limits the applicability of the proposed approach in real-world environments and generic tasks.

-Being mostly applied in simulation and pre-segmented object point cloud data, the resilience of the proposed approach to sensor noise, segmentation errors, partial observability etc is not evaluated. These factors can hinder its applicability to the real world. Additionally, the need for a segmented object pointcloud contradicts the motivation of manipulation arbitrary (previously unseen) objects.

Additionally, In my opinion, the contributions of the proposed methodology are best suited to the robotics community, where the proposed framework can be most appreciated. In the context of an ML-venue, the representation learning-related contribution appears minimal.

**Questions:**

-How can the proposed method be extended to deal with the presence of other external forces on the manipulated object?

-How can the proposed IK solution be extended to address collisions with other objects in the manipulation environment?

-How resilient is the proposed approach to RGB-D sensor noise and segmentation errors / partial observability of objects and occlusions.


Presentation:
-Many of the figures are not reference in the text, which can disturb the flow.

Notes:
-The provided supplementary material file is identical to the main submission, on the reviewer's side.
-The manuscript makes several references to an anonymised website where supplementary information is stored, which at the moment is empty of content.

Typos:
Sec.2: "Fricion Cone" -> Friction Cone
Sec.3: "Fig.??"
Sec.3:  Fig missing caption.
Sec.6: "The proposed framework." (phrase cut (?)).


Post-Rebuttal Edit:  Score reduced from 5(BR) to 3(R), given the lack of a rebuttal submission by the authors, which leaves many of the raised concerns unaddressed.

---

> ### Comment · Reviewer_G6eg · 2023-11-22
>
> Dear authors,
>
>   I believe that some of the questions raised above are crucial to verify the applicability of the proposed approach and better understand its limitations. As such, I strongly encourage you to participate in the discussion, as your insights are necessary to allow me to preserve (or increase) my score.
>
> Looking forward to your reply.

---

### Official Review · Reviewer_VBDw · 2023-11-01

**Soundness:** 2 fair
**Presentation:** 1 poor
**Contribution:** 2 fair
**Rating:** 3
**Confidence:** 4

**Summary:**

This paper introduces, UniContact, which is aiming for contact synthesis for robotic manipulationn. It addresses the challenge of choosing contact positions and forces on objects to produce specified wrenches, using point cloud segmentation and deep neural networks. The framework optimizes contact points and forces and generates collision-free joint values for manipulators. The approach is validated through both simulations and real-world experiments.

**Strengths:**

The authors have incorporated wrench consideration into the generation of contact points. They attempted a real-robot experiment, although they only provided a setup figure.

**Weaknesses:**

This paper seems to be incomplete; the provided website is empty, and the supplementary PDF is the main paper.

The writing of this paper also needs to be polished. For example, the sentence "Based on the predefined number of contact points K, ..., . Here each contact region set contains K regions. Each region contains one contact point, thus one region may appear more than once inside the contact region set." is unclear and hard to follow.

Furthermore, the authors did not introduce what clustering method was used. And if I understand correctly, clustering divides the point cloud into a finite number of regions. From these regions, K regions can be sampled to form a candidate region set (repeated sample is allowed).

"We gather the segmented point cloud leveraging the image segmentation from SAM Kirillov et al. (2023)." This is unclear. It seems that SAM can only segment 2D image, it seems the authors use 2D masks to obtain the object's point cloud, then use clustering methods to divide it into different regions. Here's a further question: the point cloud obtained from an RGB-D camera will be incomplete, how did the authors address this?

There are no particularly novel ideas in the approach, and the training data was also generated through a sampling-based method, make it less interesting.

**Questions:**

Typo: upper and lower limitations. Figure ?? on top of Page 7.
Why are there 2n+1 configurations? If for joint i, it has two possibilities L_i and H_i, while the rest are M_j, it seems there would be a total of 2n configurations.

---

### Meta-Review · Area_Chair_1Suj · 2023-12-14

**Metareview:**

This manuscript proposes an approach for generating grasp poses given a URDF of the robot hand used, a point cloud of the object, and the target wrenches. This is a challenging but important task, and a meaningful contribution in this space would certainly be of value for the community.

Unfortunately, all the reviewers (and myself) agree that the current manuscript is an incomplete work with several crucial issues. I here highlight the experimental results that are simply insufficient to support the claims of the manuscript, with a single result presented (Table 1), and linking to the (empty) website for further results.

In addition, the authors did not participate the in the rebuttal.

**Justification For Why Not Higher Score:**

The manuscript is clearly incomplete, lacking of experimental results, and the authors did not participate in the rebuttal.

**Justification For Why Not Lower Score:**

N/A

---

### Decision · Program_Chairs · 2024-01-16

Reject